# Dose-Dependent Effects of Dietary Xylooligosaccharides Supplementation on Microbiota, Fermentation and Metabolism in Healthy Adult Cats

**DOI:** 10.3390/molecules25215030

**Published:** 2020-10-29

**Authors:** Yang Lyu, Sandra Debevere, Hermann Bourgeois, Mavis Ran, Bart J.G. Broeckx, Lynn Vanhaecke, Tom Van de Wiele, Myriam Hesta

**Affiliations:** 1Department of Nutrition, Genetics and Ethology, Faculty of Veterinary Medicine, Ghent University, Heidestraat 19, 9820 Merelbeke, Belgium; yang.lyu@ugent.be (Y.L.); sandra.debevere@ugent.be (S.D.); bart.broeckx@ugent.be (B.J.G.B.); 2Longlive Biotechnology Co. Ltd., Haier Road 63, Qingdao 266103, China; hfmbourgeois@gmail.com (H.B.); mavis_ran@hotmail.com (M.R.); 3Lab of Chemical Analysis, Faculty of Veterinary Medicine, Ghent University, Salisburylaan 133, 9820 Merelbeke, Belgium; lynn.vanhaecke@ugent.be; 4Center for Microbial Ecology and Technology (CMET), Faculty of Bioscience Engineering, Ghent University, Coupure Links 653, 9000 Ghent, Belgium; tom.vandewiele@ugent.be

**Keywords:** xylooligosaccharides (XOS), microbiome, gut health, prebiotics, cats

## Abstract

In order to investigate the effect and appropriate dose of prebiotics, this study evaluated the effect of two levels of xylooligosaccharides (XOS) in cats. Twenty-four healthy adult cats were divided into three groups: no-XOS control diet with 1% cellulose; low XOS supplementation (LXOS) with 0.04% XOS and 0.96% cellulose; and high XOS supplementation (HXOS) with 0.40% XOS and 0.60% cellulose. Both XOS groups increased blood 3-hydroxybutyryl carnitine levels and decreased hexadecanedioyl carnitine levels. Both XOS treatments displayed an increased bacterial abundance of *Blautia*, *Clostridium* XI, and *Collinsella* and a decreased abundance of *Megasphaera* and *Bifidobacterium*. LXOS groups increased fecal pH and bacterial abundance of *Streptococcus* and *Lactobacillus*, decreased blood glutaryl carnitine concentration, and *Catenibacterium* abundance. HXOS group showed a more distinct microbiome profile and higher species richness, and an increased bacterial abundance of *Subdoligranulum*, *Ruminococcaceae* genus (unassigned genus), *Erysipelotrichaceae* genus, and *Lachnospiraceae*. Correlations between bacterial abundances and blood and fecal parameters were also observed. In conclusion, XOS could benefit feline gut health by altering microbiota; its effects dependant on the dose. The higher-dose XOS increased bacterial populations that possibly promoted intestinal fermentation, while the lower dose altered populations of carbohydrate-metabolic microbiota and possibly modulated host metabolism. Low-dose prebiotics may become a trend in future studies.

## 1. Introduction

Fermentation of prebiotics results in the production of short-chain fatty acids (SCFAs; acetate, propionate, and butyrate), which are generally believed to be beneficial for the host [1]. Human studies have demonstrated that prebiotics can potentially prevent a wide variety of conditions such as obesity, allergy, and inflammatory diseases, amongst others [2,3,4]. Many studies have already investigated different types of prebiotics and have demonstrated a variety of beneficial effects of those, including the improvement of insulin sensitivity in dogs and horses (short-chain fructooligosaccharides) [5,6]; the amelioration of the immune response in piglets (mannan oligosaccharide) [7]; and the better performance in poultry (lactulose) [8].

Xylooligosaccharides (XOS) are polymers of the sugar xylose; they are produced from the xylan fraction in plant fiber [9]. As a class of prebiotics, XOS has been demonstrated to exhibit a great number of health benefits for humans and rodents [10], such as the improvement of bowel function and mineral absorption, the regulation of intestinal microbiota, the reduction in colon cancer risk as well as the enhancement of antioxidant capacity, anti-inflammatory and antimicrobial functions [11,12]. Furthermore, XOS has also been reported to be effective in regulating lipid metabolism, for e.g., reducing the cholesterol and triglycerides in diabetic rats [13,14], whereas Sheu et al. noted an improvement in blood glucose and serum lipids in patients with type 2 diabetes mellitus [15]. Studies in adult humans show that the effective daily dose of XOS is only 1.4 g/day as compared to fructooligosaccharides and galactooligosaccharides, which have a minimum effective dose of 10.0 g/day [16,17,18]. This makes it possible to use lower doses of prebiotics so that side effects (e.g., excessive flatus and borborygmi) and production costs could be reduced. Moreover, its prominent stability to acid and heat compared to other indigestible polysaccharides makes it a promising supplement for pet food [19].

Despite a short colon and the lack of a functional caecum as evolutionary adaptations to a strictly carnivorous diet, considerable microbial fermentation occurs in the hindgut of domestic cats [20]. Microbiota of domestic cats have been demonstrated to be capable of fermenting a broad range of dietary fibers, including prebiotics [21]. The majority of the information on the benefit of fibers or prebiotics is, however, extrapolated from human nutrition. Dietary and metabolic interspecies differences should be considered, as domestic cats, which are obligate carnivores, rely on high protein (HP)-containing animal tissues to meet their unique nutritional requirements and are metabolically adapted to lower glucose utilization and higher protein metabolism [22,23]. In order to explore the effect and appropriate dosage of XOS, this study was established to evaluate the effects of two XOS supplementation levels in cats. The antioxidant status, fermentation pattern, metabolic profile, and microbial composition were sequentially determined in 24 healthy spayed adult cats.

## 2. Results

During the trial, body weight and body condition did not differ among the three groups. All diets were well tolerated and did not markedly affect food intake.

### 2.1. Fecal Parameters

The effect of dietary XOS supplementation on fecal parameters (fecal score, pH and fermentation pattern) is summarised in Figure 1. Compared with the control group, the low XOS supplementation (LXOS) group had significantly increased fecal pH. There was no significant effect on fecal parameters between two periods, and no diet x period interaction was observed. Data without significant differences or trends are summarised in Appendix A.

### 2.2. Blood Parameters

The effect of XOS supplementation on blood concentration of 3OH-C4, C5DC, and C16DC is shown in Figure 2. Compared with the control group, both XOS groups significantly increased 3OH-C4 concentration and decreased C16DC concentration; the LXOS group had decreased C5DC concentration in blood. No period effect and diet x period interaction was observed on blood parameters. There was no statistical difference or trends in amino acids profiles (Appendix A); data without significant differences or trends in acylcarnitine profiles are summarised in Appendix A.

### 2.3. Microbiome Profile

A total of 367,082 reads were obtained from the fecal samples in the three groups, with 96,089 reads from the control group, 128,427 from the LXOS group, and 142,566 from the high XOS supplementation (HXOS) group. The species richness in HXOS group was significantly higher than that in the control group; there were also trends in the HXOS group of an increased Shannon index and inverse Simpson index (Table 1).

A heat map with a dimensional cluster dendrogram is presented in Figure 3. From the cluster dendrogram, it is clear that XOS groups did not cluster, indicating a strong animal variability that bigger than the possible differences induced by the dietary treatment. Nevertheless, cats fed with 0.4% XOS seem to have a more distinct microbiome profile than cats fed with 0.04% XOS or the control diet. Cats in the control and LXOS groups seemed to have a similar fecal microbiome profile, as they clustered more together.

The relative abundance of the OTUs (operational taxonomic units) is summarized in Table 2. There were 16 OTUs with a significant difference in the fecal microbiome. Briefly, compared with the control group, both of XOS groups significantly increased the abundance of the genera *Blautia*, *Clostridium* XI, and *Collinsella* and decreased the abundance of *Megasphaera* and *Bifidobacterium* (*p* < 0.05); LXOS group increased the abundance of *Streptococcus*, *Lactobacillus,* and decreased the abundance of *Catenibacterium* (*p* < 0.05) while HXOS group increased the abundance of *Ruminococcaceae* genus (an unassigned genus that belongs to the family *Ruminococcaceae*), *Erysipelotrichaceae* genus, *Subdoligranulum*, and *Lachnospiraceae* genus (*p* < 0.05).

To reveal the correlations between microbiota, intestinal fermentation and body metabolism, a Spearman’s correlation analysis was performed, and the result is presented in Figure 4. Briefly, *Bifidobacterium* (OTU 2 and 27) were positively correlated with butyric acid, isobutyrate, isovalerate, and total branched-chain fatty acids (BCFA), negatively correlated with indol and C5:1; *Lactobacillus* (OTU 1, 9 and 18) were positively correlated with butyric acid, indol, C2, C5:1, and Met, negatively correlated with C4, C5:1, Fen, and Ala; *Catenibacterium* (OTU19) was positively correlated with Cit and negatively correlated with C5:1; *Blautia* (OTU 3 and 12) were positively correlated with indol, butyric acid, total SCFA, C5:1, and 3OH-C4; *Ruminococcaceae* genus (OTU 16 and 30) were positively correlated with butyric acid, isobutyrate, isovalerate, ammonia, total BCFA Orn, Cit and 3OH-C4, negatively correlated with phenol; *Collinsella* (OTU 7 and 10) were positively correlated with indol, acetic acid, isobutyrate, isovalerate, total BCFA, Orn, Cit C0, C5:1 and 3OH-C4; *Subdoligranulum* (OTU23) was positively correlated with indol, isobutyrate, isovalerate, total BCFA, C5:1 and 3OH-C4; *Lachnospiraceae* genus (OTU 15 and 24) were positively correlated with indol, propionic acid, Orn, Fen, C0, C5:1 and 3OH-C4; *Erysipelotrichaceae* genus (OTU14) were positively correlated with Orn, Cit and 3OH-C4; *Streptococcus* (OTU5) were positively correlated with C5DC; *Clostridium* XI (OTU4) were positively correlated with indol, phenol, C5:1 and Fen, negatively correlated with butyric acid.

## 3. Discussion

Many studies have investigated the effects of multiple types of prebiotics in cats, such as fructooligosaccharides (FOS) and galactooligosaccharides, using dosage ranging from 0.5% to 6% [24,25,26,27,28,29,30,31]. However, the present study showed a different effect, depending on the dose of xylooligosaccharides supplementation. Both XOS treatments influenced the fecal microbiota populations, but the effect of the higher dose was mainly associated with promoting intestinal fermentation, whereas the lower dose was mainly associated with enhancing carbohydrate metabolism.

### 3.1. Both XOS Treatments Influenced Intestinal Metabolism and Microbiome

In this study, both XOS supplemented groups showed a reduction in blood concentrations of 3OH-C4 and C16DC. The metabolite 3OH-C4 may play an important role in insulin resistance and type 2 diabetes mellitus, as shown in mice [32]. Findings in humans suggested that C16DC was deficient in fibroblasts from patients lacking functional peroxisomes and normal in fibroblasts from patients with a mitochondrial fatty acid oxidation defect [33]. Despite the lack of previous feline data, the present study shows a potential for metabolic regulation in cats by XOS supplementation.

Both XOS groups displayed a lower abundance of *Megasphaera* and *Bifidobacterium* and a higher abundance of genera *Blautia*, *Clostridium* XI, and *Collinsella*. The abundance of *Bifidobacterium* and *Megasphaera* are often upregulated in humans or dogs with increased carbohydrate fermentation, whereas the abundance of *Clostridium* XI are often downregulated in these cases [29,33,34]. Findings in this study showed an opposite shift of these bacteria compared to studies in humans or dogs. However, the role of a single bacterium might differ among the gastrointestinal microenvironments from various hosts. For instance, *Bifidobacterium* has been suggested not to be so important for cat health as for dogs or humans [35]. *Blautia* belongs to the family of *Lachnospiraceae,* and herbivores commonly harbor a higher abundance of *Lachnospiraceae* than carnivores [36]. On the other hand, studies demonstrated that *Collinsella* could modify host bile acids, alter intestinal cholesterol absorption, decrease glycogenesis, and increase triglyceride synthesis [37,38]. An increased *Collinsella* abundance in this study may demonstrate a likely association between XOS supplementation and lipid metabolism. Notably, positive correlations between *Collinsella* and several metabolites and fermentation products were also shown in the present study, indicating a potential for metabolic regulation by microbiota induced by prebiotic supplementation. These results suggest that XOS supplementation can influence the environment and ecosystem in the feline gut, but future studies are warranted to unravel underlying mechanisms and specify potential implications.

### 3.2. High XOS Supplementation could Promote Intestinal Fermentation

HXOS group had higher species richness and a more distinct microbiome profile than LXOS or the control group. Species richness promotes stability of microbiota upon increased dietary fiber intake [39], which is a positive effect of 0.4% XOS supplementation. Additionally, HXOS group increased the abundance of unclassified *Ruminococcaceae* genus, *Erysipelotrichaceae*, and *Subdoligranulum*. Increasing abundance of *Erysipelotrichaceae* was associated with diet-induced obesity by a high-fat diet in mice [40,41], yet its fluctuation by prebiotics was previously not yet investigated. The cats in the present study were lean both at the start and end of the study. *Subdoligranulum* belongs to the family *Ruminococcaceae,* which are common bacteria in the gut that help the host degrade complex carbohydrate-like prebiotics [42]. Moreover, *Subdoligranulum* and *Ruminococcaceae* showed positive correlations with SCFAs and BCFAs, emphasizing the interaction between XOS, microbiota shifts, and intestinal fermentation. These results suggested that a 0.4% XOS supplementation could alter bacterial populations that possibly promote intestinal fermentation.

### 3.3. Low XOS Supplementation could Enhance Carbohydrate Metabolism

In addition to the reduction of 3OH-C4 and C16DC, the LXOS group also decreased C5DC concentration in blood, suggesting that dietary 0.04% XOS supplementation could affect host metabolism in cats [43]. Alterations in these parameters implied that the lower dose XOS might influence carbohydrate metabolism [32,33]. However, very few studies revealed a relationship between C5DC and any metabolic factor in cats and humans [44]. LXOS group also increased the abundance of *Streptococcus* and *Lactobacillus*, decreased the abundance of *Catenibacterium*, suggesting a beneficial effect on the regulation of intestinal ecosystem and metabolism, as *Lactobacillus* has been highlighted as a beneficial gut bacterium [45,46], and *Streptococcus* plays a prominent role in the primary carbohydrate metabolism [47]. *Catenibacterium* is a Gram-positive bacterium from the family *Erysipelotrichidae*, which has been associated with metabolic disorders in other hosts [48]. Additionally, correlations were observed between C5:1 concentration and *Lactobacillus* and *Catenibacterium* as well as between C5DC concentration and *Streptococcus*, indicating the potential connection between XOS, microbiota, and metabolism. These results suggested that a 0.04% XOS supplementation could alter microbiota associated with carbohydrate metabolism and modulate host metabolism.

### 3.4. Could a Low Dose be Better?

XOS had no negative effect on the fecal scores at both doses in this study. This is in contrast with some other types of prebiotics that sometimes show an increased fecal moisture content or soft feces [24,25,26]. The softer feces could indicate a too intensive fermentation [30], but the present study showed that even the high dose (0.4%) is safe for use in practice. Often, a decreased fecal pH is expected with prebiotic supplementation due to the increased production of SCFAs and lactic acid by carbohydrate fermentation [24]. Interestingly, an increased pH was observed in the LXOS group. However, there was no significant difference when supplementing XOS at 0.4%, suggesting that the expected pH reduction may only occur at higher doses.

For the first time, this feline study observed a dose-based variation of prebiotic effects and also showed significant differences with a dose as low as 0.04%. Increased concentrations (6–9%) of fermentable fibers have led to diarrhea [49,50], while reduced concentrations (<1%) of fermentable fibers demonstrated positive effects on modulation of microbial community and body metabolism [51,52]. Notably, supplementation of 0.4% XOS was defined as a ‘high dose’ in this research, whereas this dosage is substantially lower than in other prebiotic studies. In human clinical trials, health benefits of XOS have typically been observed at 1–4 g/d, a lower dose than required for prebiotics such as fructooligosaccharides and inulin [53]. These results have raised a surprising question and hypothesis—could a low dose be better? And what is the minimal dose required to have prebiotic effects without having side-effects in terms of bloating, soft feces, or other effects? This is warranted to be explored in the future.

## 4. Materials and Methods

### 4.1. Animals and Diets

This study was approved by the Ethical Committee of the Faculty of Veterinary Medicine, Ghent University, Belgium (EC 2015/72), and was in accordance with institutional and national guidelines for the care and use of laboratory animals. Twenty-four adult domestic shorthair cats (8 males and 16 females, all neutered) with a mean age of 8.3 (standard deviation: ±2.7) years, body weight (BW) of 3.5 (standard deviation: ±0.8) kg, and body condition score (BCS, using a 9-point scale) of 4.8 (standard deviation: ±0.8) were involved in this study. After an adaptation period of three weeks, the cats were equally divided into three groups (two treatment diets and one control diet) considering an equal division of age, gender, and body weight (eight cats per group).

In order to maximize the effects of XOS, a 36% wheat-free extruded non-commercial diet without additional soluble fiber sources, based on dehydrated poultry meat and corn, was manufactured (ATM Petfood Co., Longué-Jumelles, France). Cellulose and/or XOS 35P (Longlive Biotechnology Ltd., Dezhou, China) were included in order to obtain three different isocaloric and isofibrous diets: LXOS, with a low concentration of XOS (0.04% XOS and 0.96% cellulose); HXOS, high in XOS (0.40% XOS and 0.60% cellulose) and control, XOS free diet with 1% of cellulose. The dietary ingredients and nutrient composition are depicted in Table 3.

All cats were initially adapted to the control diet (1% cellulose) for three weeks and were fed individually once a day (at 8:00 AM). The amounts of food were calculated according to their individual maintenance energy requirement (MER) by using the MER for adult spayed cats (313.6 kJ/kg 0.67) as a starting point and adapted until a stable body weight was reached according to weekly weighing and performing BCS of the cats [54]. Freshwater was continuously provided by automatic drinking fountains and bowls; both were refreshed daily. After the adaptation period, the cats were equally distributed in three treatment groups (*n* = 8) and fed once a day their assigned diet according to their adjusted MER during a period of four weeks. The trial was carried out in two periods (12 cats each period), considering an equal distribution of the different treatment groups over the two periods.

### 4.2. Sample Collection

In the last week of the test period, samples were collected. Before feeding, 8 mL of blood was aseptically drawn from the jugular vein. Immediately after collection, blood samples were transferred to Vacutainer^®^ tubes (BD Inc., Franklin Lakes, NJ, USA) containing a serum clot activator and lithium heparin. Serum and plasma were separated by centrifugation (3722× *g* for 5 min) and frozen (−20 °C). After blood collection, cats were individually housed with an empty litter box until defecation. To train the cats to use an empty litter box, litter boxes with gradually decreased amounts of litter were provided during the adaptation period. Fresh fecal samples were collected within 15 min after defecation. The samples were scored for fecal consistency (1 = watery liquid that can be poured; 2 = soft, unformed stool that assumes the shape of the recipient; 3 = soft, formed, moist stool; 4 = hard, formed, dry stool; 5 = hard, dry), fecal pH was measured with a portable pH meter (Hanna Instruments Ltd., Temse, Belgium) and samples were immediately frozen thereafter and stored at −20 °C until further analysis for fermentation markers and microbiome.

### 4.3. Analytical Methods

Tested diets were subjected to proximate and total dietary fiber analysis (Table 1). Dry matter (DM) was determined by drying to a constant weight at 103 °C, crude protein by calculating from Kjeldahl nitrogen (ISO 5983-1, 2005); crude fat, by the Soxhlet method (ISO 1443, 1973); crude fiber, by acid-alkali digestion (ISO 5498, 1981) and crude ash by combustion at 550 °C. Nitrogen-free extract was calculated by deducting crude protein, crude fat, crude fiber, and crude ash on a DM basis from 100. Total dietary fiber was determined by a commercial enzymatic digestion test kit (Sigma-Aldrich Co., St. Louis, MO, USA) and corrected for protein and ash.

Fecal samples were analyzed for SCFAs, branched-chain fatty acids (BCFAs), ammonia, indole p-cresol, phenol, and microbiota. For the analysis of SCFA and BCFA, samples were thawed, and 1 g of fecal material was transferred into a centrifuge tube together with 5 mL of water/phosphoric acid/formic acid (550:10:1, *v*:*v*:*v*). Samples were shaken manually before centrifugation (15 min at 22,000 g and at 4 °C, Beckman J2-HS, Palo Alto, CA, USA), after which the supernatant was filtered through glass wool prior to GC analysis according to Andersen et al., (2014) [54]. Fecal samples for indole, phenol, and p-cresol analysis (1 g) were ground, homogenized, and spiked with 100 µl of internal standard (100 µg/mL 5-methylindole). Extraction was performed by adding 15 mL of hexane followed by 10 min of ultrasonic vibration, 10 min of rotation (130 tpm), and further ultrasonic vibration for 10 min. After clarification at 4 °C, the supernatant was decanted, and the fecal matter was extracted a second time with 15 mL of hexane and subsequent ultrasonic vibration and rotation. Supernatants were pooled and centrifuged at 14,000 *g* for 10 min at 4 °C, and 200 µL of this combined supernatant was transferred to a GC vial for analysis. All chromatographic analyses were performed with a trace gas chromatograph coupled to a Polaris Q quadrupole ion trap mass spectrometer and a Finnigan MAT A200S autosampler (Thermo-Finnigan, Austin, TX, USA) according to Rochus et al. (2014) [20]. Ammonia nitrogen was analyzed by steam distillation according to standard methods (4500-NH3 B; APHA, 1992).

Blood was analyzed for amino acid profiles and acylcarnitines profiles. A single 3.2 mm dried blood spot (equivalent to 3.1 mL of blood) was placed in a polypropylene microtiter plate (Corning). Methanolic internal standard solution (100 mL) was added either manually or with a Labsystems Multi-drop Dispenser. The concentrations of the deuterium-labeled internal standards per litter of methanol were described by Zytkovicz et al. (2001) [55]. The microtiter plate was gently shaken during the 20-min extraction of the amino acid and acylcarnitine markers. The methanol extract was then manually transferred to a second polypropylene microtiter plate and dried by a hot air blower. Butanol-HCl (70 mL) was manually placed in each sample well, and the microtiter plate covered with a thin Teflon sheet under a heavy weight and placed in 65 °C forced air oven for 15 min. After the plate was removed from the oven, the hot air blower removed the butanol-HCl. The butanol-derivatized samples were reconstituted with 100 mL of acetonitrile and water (80:20 *v:v*), and each plate was covered with aluminum foil. The samples were then ready for MS/MS analysis, as described in [55].

For microbiota determination, fecal samples were homogenized, and total bacterial DNA was extracted according to Becker et al. (2014) [56]. Twenty-five grams (wet weight) of each thawed fecal sample was placed separately in sterile stomacher bags and homogenized in 225 mL peptone-buffered saline (PBS) (0.1% *w/v* bacteriological peptone, 0.85% *w/v* NaCl). The sludgy homogenate was filtered on a Büchner funnel to discard large particles such as hair and bones and subsequently divided into 1.5 mL aliquots, which were stored at −80 °C. The protocol of Vanhoutte et al. (2004) was used to extract total bacterial DNA from the fecal samples [57]. DNA size and integrity were assessed on 1% agarose electrophoresis gels stained with ethidium bromide. DNA concentration and purity were determined by spectrophotometric measurement at 234 nm, 260 nm, and 280 nm. Subsequently, DNA samples were sent to the company LGC Genomics (Teddington, UK) for library preparation, sequencing (Illumina Miseq), and bioinformatic analysis. 16S rRNA gene sequences were obtained from the SILVA database (Max Planck Institute for Marine Microbiology and Jacobs University, Bremen, Germany), and analysis of bioinformation was processed as described by Camarinha-Silva et al. (2014) [58].

### 4.4. Statistical Analysis

The statistical analysis was conducted in R version 4.0.2. Significance was set at α ≤ 0.05, and a *p*-value ≤ 0.1 was considered a trend. Data are reported as median and range. To assess the effect of diet, a linear model was used with diet as an independent variable and the various fecal and blood parameters as dependent variables. The significance of the dependent variable was assessed with a likelihood ratio test. When significant, a Tukey Honest Significant Differences test was conducted as a post-hoc test.

For the microbiome data, Shannon and (inverse) Simpson indexes, Pielou’s evenness, and species richness were analyzed for microbial diversity. Significant differences in bacterial abundance were evaluated using the analysis of similarities (ANOSIM). All microbiome data evaluations were performed by the online platform MicrobiomeAnalyst (McGill University, Montreal, QC, Canada). Additionally, using SPSS v24.0 software, spearman correlations coefficients were calculated between the fecal microbiome and fecal fermentation, plasma amino acids, and acylcarnitine profiles. The *p*-values were corrected with the false discovery rate method.

## 5. Conclusions

This study investigated the effect of two doses of XOS supplementation in healthy adult cats. The major finding was that both two levels of XOS affected the fecal microbiome in cats. Surprisingly, this study demonstrated a different effect, depending on the supplemented dose. The impact of the lower-dose XOS was mainly associated with carbohydrate metabolism, while the higher dose was mainly associated with intestinal fermentation. These findings indicated that supplementation XOS could benefit the gut health of cats by influencing microbiota and intestinal metabolism, while its different effects depend on the dosage. Future research is needed to further explore the association between the effects and doses of the supplementation of XOS as well as other prebiotics. Moreover, significant effects were observed even with a very low dose of XOS, pointing towards more research on the low dosage of prebiotics in the future.

## Figures and Tables

**Figure 1 molecules-25-05030-f001:**
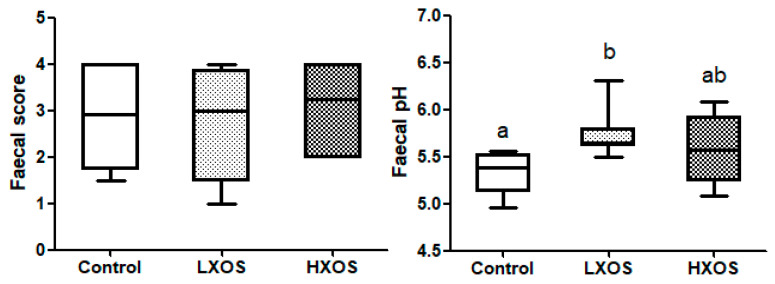
Effect of xylooligosaccharides (XOS) on fecal score and pH. Groups with different letters (a, b) are significantly different (*p* < 0.05).

**Figure 2 molecules-25-05030-f002:**
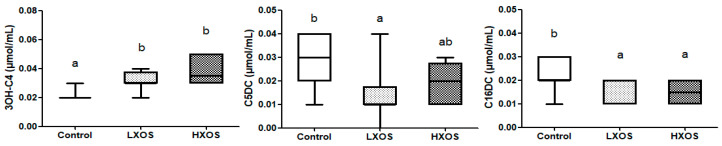
Effect of XOS on the blood concentration of 3OH-C4, C5DC, and C16DC. 3OH-C4: 3-hydroxybutyryl carnitine; C5DC: glutaryl carnitine; C16DC: hexadecanedioyl carnitine. Groups with different letters (a, b) are significantly different (*p* < 0.05).

**Figure 3 molecules-25-05030-f003:**
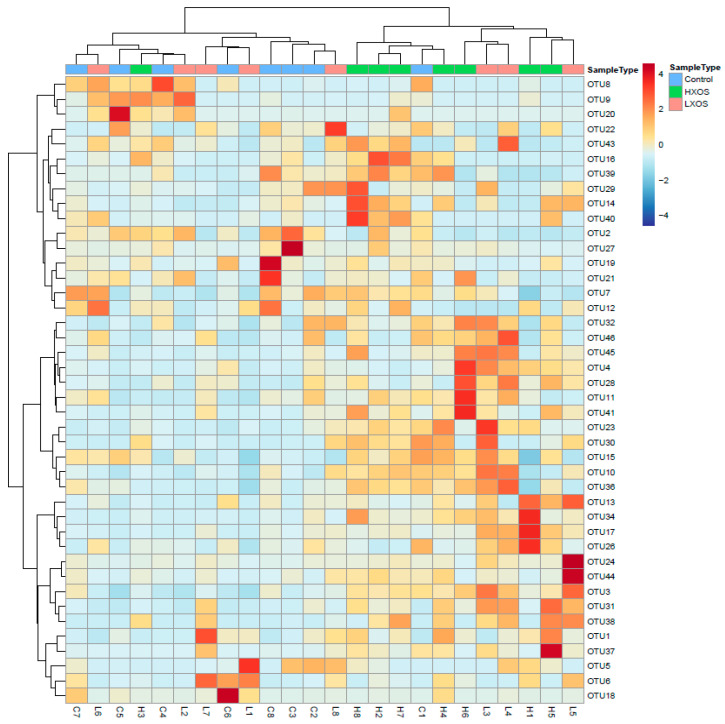
Heat map of the relative abundance and dendrogram clustering of the fecal microbiota genera. Colors with values represent log-transformed relative abundance of operational taxonomic units (OTUs).

**Figure 4 molecules-25-05030-f004:**
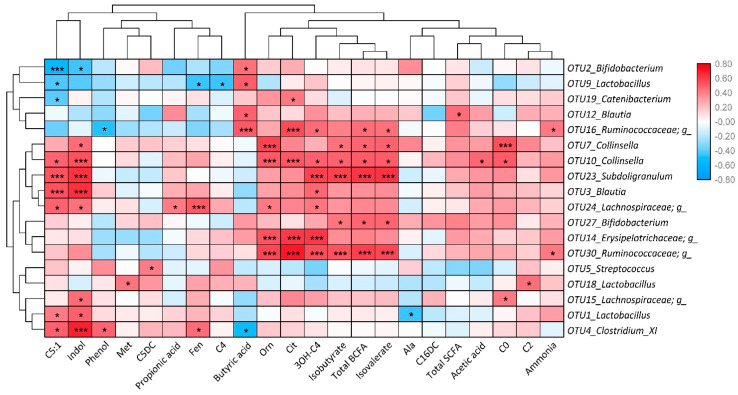
Spearman correlations between OTU abundance and fermentation and metabolism profile. Red represents a positive correlation, while blue represents a negative correlation. “g_” means unclassified genus. * *p* < 0.05, *** *p* < 0.01.

**Table 1 molecules-25-05030-t001:** Impact of XOS supplementation on diversity, richness, and evenness of the fecal microbiome.

Item.	Control	LXOS	HXOS	SEM	*p* Value
Shannon index	1.94	1.91	2.23	0.11	0.077
Simpson index	0.78	0.758	0.838	0.03	0.185
Inverse Simpson index	4.86	5.05	6.49	0.583	0.061
Pielou’s evenness	0.499	0.463	0.536	0.023	0.271
Species richness	49.4 ^a^	61.6 ^ab^	64.8 ^b^	4.41	0.022

^a,b^ values within a row with different letters differ (*p* < 0.05). LXOS: low XOS supplementation group; HXOS: high XOS supplementation group.

**Table 2 molecules-25-05030-t002:** Relative abundance of the operational taxonomic units (OTUs).

Taxonomy	Control	LXOS	HXOS	*p* Value
**Firmicutes**				
Bacilli; Lactobacillales; *Streptococcaceae*; *Streptococcus*	13.468 ^a^	28.510 ^b^	7.561 ^a^	<0.001
Bacilli; Lactobacillales; *Lactobacillaceae*; *Lactobacillus*	7.044 ^a^	16.591 ^b^	11.403 ^ab^	<0.001
Clostridia; Clostridiales; *Ruminococcaceae*; unclassified genus	1.523 ^a^	0.871 ^a^	4.522 ^b^	<0.001
Clostridia; Clostridiales; *Lachnospiraceae*; *Blautia*	0.396 ^a^	1.077 ^b^	1.870 ^c^	<0.001
Erysipelotrichia; Erysipelotrichales; *Erysipelotrichaceae*; unclassified genus	1.495 ^a^	2.193 ^a^	4.294 ^b^	0.001
Clostridia; Clostridiales; *Ruminococcaceae*; *Subdoligranulum*;	0.240 ^a^	0.398 ^a^	0.771 ^b^	0.001
Erysipelotrichia; Erysipelotrichales; *Erysipelotrichaceae*; *Catenibacterium*	1.669 ^b^	0.587 ^a^	1.178 ^ab^	0.010
Clostridia; Clostridiales; *Lachnospiraceae*; *Blautia*	2.024 ^a^	4.051 ^b^	2.085 ^a^	0.029
Clostridia; Clostridiales; *Lachnospiraceae*; *Blautia*	6.423 ^a^	12.713 ^b^	9.363 ^ab^	0.038
Clostridia; Clostridiales; *Lachnospiraceae*; unclassified genus	0.127 ^a^	0.399 ^ab^	0.505 ^b^	0.044
Negativicutes; Selenomonadales; *Veillonellaceae*; *Megasphaera*	1.337 ^b^	0.790 ^a^	0.819 ^a^	0.045
Clostridia; Clostridiales; *Peptostreptococcaceae*; *Clostridium* XI	2.198 ^a^	3.258 ^b^	3.094 ^b^	0.047
**Actinobacteria**				
Actinobacteria; Coriobacteriales; *Coriobacteriaceae*; *Collinsella*	1.705 ^a^	2.556 ^b^	4.348 ^b^	<0.001
Actinobacteria; Bifidobacteriales; *Bifidobacteriaceae*; *Bifidobacterium*	0.506 ^b^	0.177 ^a^	0.241 ^a^	<0.001
Actinobacteria; Bifidobacteriales; *Bifidobacteriaceae*; *Bifidobacterium*	14.403 ^b^	16.259 ^b^	4.466 ^a^	0.006
Actinobacteria; Bifidobacteriales; *Bifidobacteriaceae*; *Bifidobacterium*	19.854 ^b^	5.829 ^a^	7.224 ^a^	0.011

^a,b^ values within a row with different letters differ (*p* < 0.05).

**Table 3 molecules-25-05030-t003:** Ingredients and macronutrient composition of control and experimental diets.

**Ingredient (%)**	**Control**	**LXOS**	**HXOS**
Corn	39.11	39.11	39.11
Dehydrated poultry meat	19.80	19.80	19.80
Corn gluten meal	9.90	9.90	9.90
Poultry Fat	6.93	6.93	6.93
Rice	4.95	4.95	4.95
Soy	4.95	4.95	4.95
Hydrolyzed dehydrated poultry meat	4.95	4.95	4.95
Premix	2.48	2.48	2.48
Fishmeal	2.23	2.23	2.23
Liquid aroma	1.98	1.98	1.98
Yeast	1.24	1.24	1.24
Aroma powder	0.50	0.50	0.50
Cellulose	1.00	0.96	0.60
XOS	0.00	0.040	0.40
**Analysis on DM basis (%)**	**Control**	**LXOS**	**HXOS**
Crude protein	33.90	32.89	36.22
Crude fat	11.49	11.56	12.14
Crude ash	6.80	6.26	5.90
Crude fiber	1.57	1.42	1.47
Total dietary fiber	10.50	10.93	11.51
Nitrogen free extract	46.24	47.87	44.27

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
