# Peer review of "Dose-Dependent Effects of Dietary Xylooligosaccharides Supplementation on Microbiota, Fermentation and Metabolism in Healthy Adult Cats"

_molecules, 2020, doi:10.3390/molecules25215030_

Round 1

Reviewer 1 Report

The authors present the alterations in the feline gut microbiome as a response to two different concentrations of dietary Xzlosaccharides. Although the microbial abundances show a high individual variation, significant trends were identified by the authors that are of interest.

Minor comments:

Some basic English grammar errors are present in the MS which should be carefully edited. For e.g. Abstract, line no. 30: ..., XOS could benefits... should actually read ..., XOS could benefit...

For Figure 4, was the average abundance of OUTs across all individuals combined across groups used to conduct correlation analysis? If not, please provide group (L/H) segregated correlation analysis in this figure.

Author Response

Thank you so much for your comments and suggestion. We have improved the manuscript based on suggestions from you and another reviewer, please check the new version of this article.

Here are the answers to your questions:

1. Some basic English grammar errors are present in the MS which should be carefully edited. For e.g. Abstract, line no. 30: ..., XOS could benefits... should actually read ..., XOS could benefit...

Sorry for our mistakes and thanks for your notice. We have revised this error and check throughout the manuscript, please check the new version.

2. For Figure 4, was the average abundance of OUTs across all individuals combined across groups used to conduct correlation analysis? If not, please provide group (L/H) segregated correlation analysis in this figure.

Thanks for your question and advice. Figure 4 was generated based on the average OTU abundance of all samples. It is indeed very interesting to evaluate the segregated correlations between treatments, we wish this would be done in a further investigation.

Reviewer 2 Report

The research done is very good and complete. There is only a couple of concerns about the XOS and cellulose concentrations and the amount of animals used in the experimental diets. The rest of the manuscript is clear and well explained, except for the introduction and methods, which should be improved. Please find bellow the comments or suggestions.

ABSTRACT

Line 33-34: According to the final lines, why is it better to focus in low-dose prebiotics?

KEYWORDS

The keywords are good. However, to increase the visibility of your research, it is better to put different words than those find in the title. For example, you could put microbiome profile, faecal pH, bacterial abundances, gut health.

INTRODUCTION

Line 47: Where xylooligosacharides come from and why the interest of application in the experimental diet of cats?

In general, what is the hypothesis of this research:

  1. The use of xylooligosacharides and explore its effect and appropriate dosage, or,
  2. The short colon and the lack of a functional caecum and its effect on the fermentation of dietary fibers in cats?

In general, this section can be improved. There is not much information about fermentation process and its effect on microbiome in other species, which can help to correlated the information with your hypothesis.

MATERIALS AND METHODS

Line 220: Why the number of cat was 24? Please, leave it clear if this amount of animals is enough for the statistical analysis.

Line 221-222: Respect to the domestic shorthair cats, results could be affected by the sex of the cat? Or does it not have a relevance in the obtained results? Please, explain.

Line 229: Why cellulose and XOS were selected for the experimental diets? And how was the concentration of both of them selected?

Lines 257-280: According to the aims of the journal, to reproduce the results, full experimental details must be provided. So, please, it is recommended to be more specific in the details of each experiment.

Author Response

Thank you so much for your valuable comments and suggestions. We have carefully considered all your comments and improved the manuscript according to your suggestions, please check the new version of this article.

Here are the answers to each question:

1. Line 33-34: According to the final lines, why is it better to focus in low-dose prebiotics?

Thanks for your question. A high dose of prebiotics can cause side effects (e.g. excessive flatus, borborygmi), so a lower dose would be better. This point has also been added in line 58-61 of the new version.

2. Keywords

Thanks for your suggestion. We have updated the keywords, please check the new version.

3. Line 47: Where xylooligosacharides come from and why the interest of application in the experimental diet of cats?

Thanks for your question and advice. XOS are polymers of the sugar xylose, they are produced from the xylan fraction in plant fibre. The lower dose of XOS could reduce the side effects and production costs, and its prominent stability to acid and heat compared to other indigestible polysaccharides makes it a promising supplement for pet food. These have been added to the line 50-51 and 57-62 of the new version.

4. Line 220: Why the number of cat was 24? Please, leave it clear if this amount of animals is enough for the statistical analysis.

Thanks for your question. Due to ethical reason and lab condition, only 24 cats were involved in this study. However, we believe it is enough for the statistical analysis, as several other prebiotics studies in cats had n≤8 (Heata et al., 2001,2005; Verbrugghe et al., 2009; Barry et al., 2014).

5. Line 221-222: Respect to the domestic shorthair cats, results could be affected by the sex of the cat? Or does it not have a relevance in the obtained results? Please, explain.

Thanks for your question. During the study we did not observe any significant difference between the males and females, it was also difficult to evaluate the effect as the power was too low. However, it is indeed a very interesting topic, we wish this would be evaluated in the future investigations. 

6. Line 229: Why cellulose and XOS were selected for the experimental diets? And how was the concentration of both of them selected?

Thanks for your question. XOS was selected because of its beneficial effects and lower used dose as discussed above, and cellulose was selected as a negative control as it is a insoluble dietary fiber while XOS is soluble. The concentration of cellulose was selected according to the previous studies (Sunvold et al.,1995; Barry et al., 2010), and the concentration of XOS was estimated according to the results in human and rodents (Imaizumi et al., 1991; Gobinath et al., 2010; Broekaert et al., 2011). Also, this study was designed with two levels, it is in consideration of finding the appropriate dose.

7. Lines 257-280: According to the aims of the journal, to reproduce the results, full experimental details must be provided. So, please, it is recommended to be more specific in the details of each experiment.

Thanks so much for your suggestions. We have added detailed information of methods and materials, please check the new version.